# Verification of the Japanese Version of Greene's Moral Dilemma Task's Validity and Reliability

**Yoshiyuki Takimoto** [1,*] 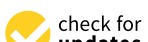 **and Akira Yasumura** [2]

1   Department of Biomedical Ethics, Faculty of Medicine, The University of Tokyo, 7-3-1 Hongo, Bunkyo-ku, Tokyo 113-0033, Japan
2   Graduate School of Humanities and Social Sciences, Kumamoto University, 2-40-1 Kurokami, Chuo-ku, Kumamoto City 860-8555, Japan
*   Correspondence: takimoto@m.u-tokyo.ac.jp

**Abstract:** The moral dilemma task developed by Greene et al., which comprises personal and impersonal moral dilemmas, is useful for clarifying people's moral judgments. This study develops and validates a Japanese version of this questionnaire. Ten new questions were added to the Japanese version using back-translation, and its internal validity was tested. A second survey was conducted among the same participants one month after the first survey ($n$ = 231). The intraclass correlation coefficient through retesting was found to be 0.781. Test-retest, internal consistency, and criterion-related validity were confirmed by retesting the Japanese version of the moral dilemma task. Moral judgments differed in gender, with women and men tending to be more utilitarian in situations where emotions were less and more likely to be involved, respectively. The association between age and deontological moral judgments was also observed.

**Keywords:** moral psychology; moral dilemma; task validation; judgment

## 1. Introduction

Over the last two decades, empirical research based on moral emotionalism has advanced the psychological elucidation of our moral judgments [1]. Accordingly, the moral dilemma issue, represented by the trolley problem [2,3], has played a pivotal role. The trolley problem asks what the person (you) standing at the junction of tracks should do in a situation where a trolley running on the tracks is out of control. If the person (you) continues, five workers will die, and if the person switches points (junctions) to save the five workers, one worker will die. This thought experiment poses the following ethical dilemma: "Is it appropriate to sacrifice one person to save many?" The typical trolley problem entails the trolley dilemma (pushing the button to divert the trolley) and the footbridge dilemma. The footbridge dilemma is a case of greater psychological resistance than the trolley dilemma because it requires pushing a person directly off a bridge. The moral dilemma task was developed based on ethical dilemmas concerning the use of consequentialism or intuitivist judgment to make decisions.

In the trolley problem, people's judgments between the trolley and footbridge dilemma differ [4]. This difference is explained by personalness or directness [5,6]. The concept of "personalness" or "directness" in a moral dilemma involves (a) severe physical harm, (b) to one or more specific individuals, and (c) when it is not the result of averting an existing threat [6] and personal force; for example, pushing an overweight man off the bridge. The impersonal moral dilemma involves indirect action unlikely to entail any emotion; for example, pressing the switch to divert the trolley.

Greene et al. [7] used functional Magnetic Resonance Imaging (fMRI) to evaluate brain function in a moral dilemma task and explained the difference in judgment between personal and impersonal moral dilemmas in terms of the emotional responses of the brain. The dual-process theory has been applied to explain the difference in utilitarian

or non-utilitarian judgment between moral personal and impersonal dilemmas. In this theory, the moral dilemma judgement is explained by the competition between automatic (ventral) and control (dorsal) nervous system reactions. Utilitarian judgment is related to controlled nervous system reactions, whereas non-utilitarian judgment is related to automatic reactions [8]. Dual-process theorists generally agree that System 1 processes are rapid, parallel, and automatic; only their final product is posted in the consciousness. System 2 is considered to have evolved more recently; most theorists further consider it uniquely human. System 2 thinking is slow and sequential and uses the central working memory system, which has been comprehensively examined in the psychology of memory. Despite its limited capacity and slower speed of operation, System 2 allows for abstract hypothetical thinking that cannot be achieved by System 1 [9]. Hence, personal moral dilemma problems are considered strongly influenced by System 1 because they tend to evoke emotions automatically.

The model contends that moral judgment is a product of two partially separable neural systems—one is fast, automatic, and effective, and the other is slower, effortful, and more abstract [5,6]. This dual process model theory, proposed by Greene et al. in 2001 [7], has been extensively examined in different cultures, including Spanish [10], Chinese [11], Italian [12], and Korean [13].

The moral dilemma task used in Greene et al.'s [7] study comprised 64 questions categorized as moral personal, moral impersonal, and nonmoral. It entailed trolley problems in the trolley and footbridge dilemma. Studies adopting brain function measures, those related to the behavior of moral judgments, and those of ethical judgments in medicine, have used it extensively [14–16]. Although Greene et al.'s [7] moral dilemma task is a useful measure of personalness or directness, it has not been tested for reliability or validity.

The relationship between personalness or directness and emotional responses, and the moral judgments based on them show that age and gender are cross-cultural [17]. Cushman et al. [4] conducted a large web-based survey with more than 5000 respondents and found that education level or religious background insignificantly affect the tendency in the dilemma task. A recent paper examining Greene et al.'s [7] moral dilemma task in 45 countries showed no relationship between cultural background differences, such as collectivism and individualism, and differences in utilitarian judgments [18]. Although the moral dilemma task involving harm aversion, which enhances the tendency to avoid utilitarian judgments in a moral personal dilemma, is reported to be cross-cultural, the effects of gender, age, and ethnic identity are notable [19,20]. Previous research shows that women scored higher than men on deontological tendencies [19]. Rechek et al. [21] found that factors such as age, gender, genetic relatedness, and potential reproductive opportunity influence moral judgments. Their results showed that being younger, having genetic relatedness, and having a lover reduced utilitarian judgments.

Accordingly, while the task proposed by Greene et al. [7] is a useful moral dilemma task that can be considered cross-culturally applicable, there are issues that have not been validated. Therefore, this study aims to develop a Japanese version of the moral dilemma task and examine its reliability and validity. Furthermore, we investigate the differences in utilitarian or non-utilitarian judgments among Japanese speakers, add knowledge about linguistic universality to the findings of Cushman et al. [4], and investigate the differences in gender and age between utilitarian and non-utilitarian judgments. Additionally, the reactions to the moral dilemma task have been compared across regions because cross-cultural differences have not been identified across regional cultural differences within Japan. Although this study's significance is limited because it does not directly verify the reliability and validity of the English version of the moral dilemma task, verifying the reliability and validity of the Japanese version will indirectly contribute to the verification of the original English version.

## 2. Materials and Methods

### 2.1. Overview of the Moral Dilemma Questionnaire

The moral dilemma questionnaire [7] comprises 64 questions and is divided into three categories. The first category comprises (a) the personal moral dilemma task (MP, e.g., the footbridge problem), which involves physical contact or other direct sacrifices of the few to help the many; (b) the impersonal moral dilemma task (MI, e.g., the trolley problem), which involves an indirect sacrifice of the few to help the many; and (c) the MI (e.g., the trolley problem), which involves an indirect sacrifice of the many to help the few. Altogether, these three subcategories comprise 25 questions. The second category comprises impersonal moral dilemmas (e.g., the trolley problem), which involve indirectly helping the many at the expense of the few (19 questions), and the third category comprises nonmoral issues (20 questions). The response format was a four-point scale:1 = extremely inappropriate, 2 = inappropriate, 3 = appropriate, and 4 = extremely appropriate. Additionally, there were 19 reversed items.

### 2.2. Procedure for Developing the Japanese Version of the Moral Dilemma Problem

The developers of the moral dilemma task approved the development of the Japanese version of the same. First, the authors translated the questions of the moral dilemma task into Japanese, after which a native speaker unaware of the original text back-translated the questions. While comparing the back-translation with the original text, the authors, who are experts in ethics, revised the Japanese version. During the two-month period, four meetings were held regarding the revision process; simultaneously, individual work was being carried out. Furthermore, a native speaker unaware of the original text performed a back-translation to confirm the revised Japanese version.

The results of the second back-translation were presented to the developers, who confirmed the items as a whole, made corrections based on the comments on items with problematic expressions and gave their final approval of the Japanese version (Supplementary Materials).

### 2.3. Modification and Addition of Items

As instructed by the developers, three low-conflict questions (Q44, Q47, and Q53 in the original version) were deleted because they were not effective in distinguishing between personal and impersonal dilemmas in previous studies [6,14]. One question was deleted owing to cultural differences between schools in Japan. Consequently, there were 22 questions for personal moral dilemma tasks and 19 questions each for tasks related to the impersonal moral dilemma and nonmoral issues. Furthermore, 10 questions from the impersonal moral dilemma (Q30–39) and nonmoral-related tasks (Q11–20) were each set as reversed items, similar to the original version. For validation, four questions of MPs and MIs each (Q65–72) (impersonal tasks were reversed items), created under the trolley problem, were added as additional items only in the second survey.

### 2.4. Survey Participants and Methods

A total of 364 people who read the explanation of the study and agreed to participate registered with the research company Macromill Inc., Tokyo, Japan (https://group.macromill.com/ (accessed on 1 February 2023)). The sample size was determined by referring to previous studies on questionnaire development [22–24]. The selection criteria included adults with Japanese nationality and no disabilities, such as mental illness. To eliminate regional differences, equal participant distribution from all regions of Japan was ensured. A web survey was conducted by Macromill, Inc. To verify the reliability of the survey, a second survey was conducted among the participants, one month after the first survey. The survey was conducted anonymously so that the participants could not be identified.

Participants who participated only in one web survey were excluded from the analysis ($n$ = 64). Those with a small variation in the answers to all questions (standard deviation

of 0.2 or less) (*n* = 3) and whose time required to respond was less than 1/16th or more than 1/16th of the total were excluded from the analysis (*n* = 69) because they were likely to have answered the questions without appropriate reflection. Therefore, a total of 231 remaining cases were considered valid responses.

*2.5. Analysis Method*

To verify the reliability, Cronbach's alpha coefficient was obtained as an index of internal consistency, and intraclass correlation coefficients (ICC) were calculated as a measure of test-retest reliability. Moreover, the criterion-related validity was calculated using Greene et al.'s [7] moral dilemma tasks and MI, and those of additional moral dilemma tasks and MI of the trolley problem. Furthermore, to examine differences by region, an analysis of variance was conducted on regional differences in the first category. To examine differences by gender, the means in the first category were compared between men and women using a t-test. To examine differences by development, correlations between means by first category and age were examined. All analyses were two-tailed, and a *p*-value of < 0.05 was considered statistically significant.

*2.6. Compliance with Ethical Standards*

This study was approved by the Ethics Committee (No. 11468-1, 14 April 2017). All the participants provided informed consent.

## 3. Results

*3.1. Participants' Characteristics*

The participants included 120 men and 111 women. The mean age was 40.2 years (SD ± 11.3, range 20.0–59.0) (Table 1).

**Table 1.** Participants' characteristics.

| Gender | (*N*) | NM | | MI | | MP | |
|---|---|---|---|---|---|---|---|
| | | Mean | *SD* | Mean | *SD* | Mean | *SD* |
| Male | 120 | 1.804 | 0.401 | 1.858 | 0.330 | 1.926 | 0.464 |
| Female | 111 | 1.723 | 0.398 | 1.948 | 0.282 | 1.652 | 0.353 |

| Regions | (*N*) | NM | | MI | | MP | |
|---|---|---|---|---|---|---|---|
| | | mean | *SD* | mean | *SD* | mean | *SD* |
| Hokkaido | 29 | 1.738 | 0.430 | 1.895 | 0.318 | 1.814 | 0.403 |
| Tohoku | 25 | 1.758 | 0.374 | 1.895 | 0.225 | 1.760 | 0.329 |
| Kanto | 35 | 1.817 | 0.403 | 1.943 | 0.354 | 1.833 | 0.410 |
| Chubu | 34 | 1.663 | 0.359 | 1.853 | 0.278 | 1.781 | 0.370 |
| Kinki | 25 | 1.798 | 0.472 | 1.853 | 0.314 | 1.816 | 0.631 |
| Chugoku | 31 | 1.682 | 0.400 | 1.890 | 0.293 | 1.694 | 0.519 |
| Shikoku | 27 | 1.780 | 0.358 | 1.890 | 0.295 | 1.796 | 0.443 |
| Kyushu, Okinawa | 25 | 1.922 | 0.396 | 2.004 | 0.391 | 1.875 | 0.352 |

| Age | Years | | | | | | |
|---|---|---|---|---|---|---|---|
| Mean | 40.2 | | | | | | |
| *SD* | 11.3 | | | | | | |
| Range | 20–59 | | | | | | |

Note. NM = nonmoral issue; MI = impersonal moral dilemma task; MP = personal moral dilemma task.

*3.2. Reliability and Validity*

The reliability of the instrument was verified using a test-retest method. Cronbach's alpha coefficient for all items in the first survey was 0.821 and that in the second survey was 0.885 (excluding the additional items). The ICC (1, 2) for all items was 0.795 (95% CI: 0.735, 0.842) (Table 2).

**Table 2.** Internal consistency and reliability using retest methods.

| | First Survey | Second Survey | |
| --- | --- | --- | --- |
| | Cronbach's α | Cronbach's α | ICC (1, 2) (95% CI) |
| NM (Q1–20; Q14 excluded) (*n* = 231) | 0.791 | 0.824 | 0.794 (0.733, 0.841) |
| MI (Q21–39) (*n* = 231) | 0.624 | 0.677 | 0.722 (0.640, 0.785) |
| MP (Q40–64; Q44, 47, 53 excluded) (*n* = 231) | 0.876 | 0.910 | 0.850 (0.806, 0.884) |
| ALL (Q1–64; Q14 & Q44, 47, 53 excluded) (*n* = 231) | 0.823 | 0.864 | 0.795 (0.735, 0.842) |

Note. NM = nonmoral issue; MI = impersonal moral dilemma task; MP = personal moral dilemma task.

The criterion validity testing showed that the first survey in the personal moral dilemma task and the footbridge dilemma task of the additionally created trolley dilemma were significantly positively correlated (r = 0.653, *p* < 0.001). A weak positive correlation was found for the MI (r = 0.226, *p* < 0.001). The mean scores of all items from the first and second surveys were plotted (Figure 1).

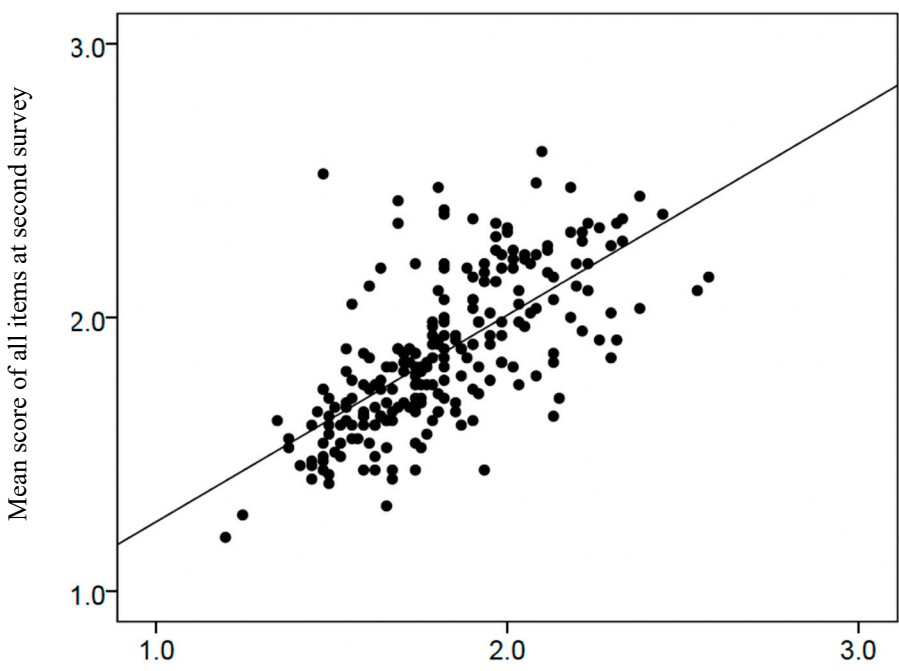

**Figure 1.** Correlation between the mean scores of the first and second surveys for all dilemma task items.

### 3.3. Greene et al.'s Dilemma Task

The mean scores of the impersonal moral dilemma, personal moral dilemma, and nonmoral tasks were 1.90 (SD = 0.31, range 1.05–2.68), 1.79 (SD = 0.44, range 1.00–3.18), and 1.76 (SD = 0.40, range 1.00–2.65), respectively.

### 3.4. Trolley Dilemma and Footbridge Task

The mean scores of the trolley and footbridge dilemmas in Greene's dilemma task were 2.53 (SD = 0.97) and 1.55 (SD = 0.78), respectively (Table 3). The ICC values (1, 2) for the trolley and footbridge dilemmas were 0.55 (95% CI: 0.417, 0.653) and 0.576 (95% CI: 0.450, 0.673), respectively.

**Table 3.** Result of trolley (Q21) and footbridge (Q41) case.

| | Non-Utilitarian | | → | Utilitarian | | | | Test-Retest | Criterion Validity Test |
| | 1 | 2 | 3 | 4 | Mean | (SD) | ICC (1, 2) (95% CI) | r (p Value) |
|---|---|---|---|---|---|---|---|---|---|
| Trolley (n = 231) | 35 (15.2%) | 83 (35.9%) | 69 (29.9%) | 44 (19.0%) | 2.53 | 0.97 | 0.550 (0.417, 0.653) | −0.679 (p < 0.0001) |
| Footbridge (n = 231) | 134 (60.6%) | 61 (26.4%) | 24 (10.4%) | 12 (2.6%) | 1.55 | 0.78 | 0.576 (0.450, 0.673) | 0.641 (p < 0.0001) |

Note. → = from non-utilitarian to utilitarian tendency.

### 3.5. Differences by Region

No regional differences were noted in any item or category (MP: $F_{(7, 223)} = 0.433$, $p = 0.881$, ηp2 = 0.029; MI: $F_{(7, 223)} = 0.699$, $p = 0.673$, ηp2 = 0.021; nonmoral related task: $F_{(7, 223)} = 1.110$, $p = 0.361$, ηp2 = 0.029).

### 3.6. Differences by Gender

The examination of gender differences showed that men were significantly more utilitarian in the MP ($t_{(229)} = 5.02$, $p < 0.001$, d = 0.61). Women were significantly more utilitarian in MIs ($t_{(229)} = -2.22$, $p = 0.027$, d = 0.29). No difference was found in nonmoral-related tasks ($t_{(229)} = 1.55$, $p = 0.122$, d = 0.20) (Table 4).

**Table 4.** Differences by gender.

| | Men (n = 120) | | Women (n = 111) | | t (df) | p | 95% CI | d |
| | Mean | (SD) | Mean | (SD) | | | | |
|---|---|---|---|---|---|---|---|---|
| NM | 1.80 | (0.40) | 1.72 | (0.40) | 1.55 (229) | 0.122 | [−0.022, 0.185] | 0.20 |
| MI | 1.86 | (0.33) | 1.95 | (0.28) | −2.22 (229) | 0.027 | [−0.170, −0.101] | 0.29 |
| MP | 1.90 | (0.46) | 1.65 | (0.35) | 5.02 (229) | <0.001 | [0.167, 0.381] | 0.61 |

Note. NM = nonmoral issue; MI = impersonal moral dilemma task; MP = personal moral dilemma task; df = degree of freedom; 95% CI = confidence intervals of mean difference; d = Cohen's d.

### 3.7. Differences by Age and Developmental Change

Participants in their 50s made significantly more utilitarian choices in the personal moral dilemma task than those in their 20s ($t_{(115)} = 2.31$, $p = 0.023$, d = 0.44) (Table 5). The examination of developmental change revealed a weak negative correlation between age and responses in the moral dilemma task overall and for men (men and women: r = −0.206, $p = 0.002$, men only: r = −0.285, $p = 0.002$) (Table 6).

**Table 5.** Differences by age.

| | 20 s (n = 53) | | 50 s (n = 64) | | t (df) | p | 95% CI | d |
| | Mean | (SD) | Mean | (SD) | | | | |
|---|---|---|---|---|---|---|---|---|
| NM | 1.80 | (0.41) | 1.73 | (0.38) | 0.83 (115) | 0.536 | [−0.849, 0.207] | 0.18 |
| MI | 1.89 | (0.33) | 1.91 | (0.29) | −0.34 (115) | 0.734 | [−0.134, 0.943] | 0.06 |
| MP | 1.87 | (0.43) | 1.69 | (0.39) | 2.31 (115) | 0.023 | [0.237, 0.328] | 0.44 |

Note. NM = nonmoral issue; MI = impersonal moral dilemma task; MP = personal moral dilemma task; df = degree of freedom; 95% CI = confidence intervals of mean difference; d = Cohen's d.

**Table 6.** Differences by developmental change.

| | Men | | Women | | Total | |
|---|---|---|---|---|---|---|
| | *r* | *p* | *R* | *p* | *r* | *p* |
| NM | 0.037 | 0.689 | −0.140 | 0.143 | −0.043 | 0.520 |
| MI | 0.138 | 0.133 | −0.134 | 0.160 | 0.011 | 0.867 |
| MP | −0.285 | 0.002 * | −0.169 | 0.077 | −0.206 | 0.002 * |

Note. NM = nonmoral issue; MI = impersonal moral dilemma task; MP = personal moral dilemma task; * $p < 0.05$ by Pearson's correlation.

## 4. Discussion

This study had two aims. First, to develop a Japanese version of Greene et al.'s [7] moral dilemma task—which is often used to measure brain functions related to moral judgments—and to test its reliability and validity. The second aim of this study was to explore whether the moral dilemma task is cross-cultural in nature and if there are gender and age-based differences in the response tendencies. To achieve this, we surveyed Japanese participants using the developed Japanese version of Greene et al.'s moral dilemma task. The results of this study demonstrated that the developed Japanese version of Greene et al.'s [7] moral dilemma task was reliable and valid. Furthermore, the findings demonstrated that the moral dilemma task was cross-cultural. Moral judgments differed based on gender, with women and men tending to be more utilitarian in situations where emotions were less and more likely to be involved, respectively. The association between age and deontological moral judgments was also observed.

A test-retest method was used to verify the reliability. The ICC values for the MI, MP, and nonmoral issue were 0.850, 0.722, and 0.794, respectively. Cronbach's alpha coefficient for MP, MI, and the nonmoral issue were 0.876, 0.624, and 0.791 for the first survey and 0.910, 0.677, and 0.824 for the second survey, respectively. All the values except Cronbach's alpha in the MI were high enough to suggest that the Japanese version of the moral dilemma task is highly reliable. The relatively low Cronbach's alpha coefficient for the MI could be because the MI is more sensitive to the cultural background than the MP.

Considering that the purpose of the study was to develop a Japanese version, we used expressions that are more intuitive to Japanese speakers. For example, "meters" instead of "yards" and "rice cutter" (Supplementary Materials).

Regarding criterion-related validity, the footbridge and trolley dilemmas in the trolley problem were used as indicators of the personal and impersonal moral dilemma tasks, respectively. A certain degree of validity was obtained for each participant. The footbridge and trolley dilemmas can be interpreted as personal or impersonal differences and also as differences in people being used as a means and the corresponding action being intentional [25]. From this perspective, further discussion is required to determine whether the footbridge and trolley dilemmas are valid external indicators of criterion-related validity.

Japanese respondents made more utilitarian moral judgments in the MI and fewer utilitarian moral judgments in the MP. The results suggest that automatic (ventral) processing as System 1 [8] was performed in the MP and control (dorsal) processing as System 2 [9] was performed in the MI, in accordance with Greene et al.'s [7] dual-processing theory. This reaffirms that the tendency toward moral judgments is cross-cultural in Japan, as noted in previous studies in other countries [10–12].

No regional differences were observed in Japan, suggesting that uniform results could be obtained across the country. This result was consistent with those reported by Hauser et al. [26] and Bago et al. [18]. Conversely, Awad et al. [27] and Gold et al. [28] reported that when distinguishing between moral judgments and behavioral intentions, regional differences were observed in behavioral intentions rather than in moral judgments. Yamamoto and Yuki [29] explain regional differences in behavioral intention reactions in relation to relational fluidity. As this study examines reactions in moral judgments, the lack of regional differences is consistent with previous studies; however, regarding behavioral intentions, the reactional differences may have been present in regions with low fluidity.

When considering the effect of the act on reputation, regional differences based on relational fluidity may be less likely to emerge in the footbridge dilemma, as only a bad reputation rather than a good one is likely to be obtained in this dilemma.

Regarding gender differences, men were more utilitarian in the MP and women were more utilitarian in the MI. Previous studies suggest that women are more deontological in personal rather than impersonal moral dilemmas [30–32]. The finding that women are more utilitarian in the MI is inconsistent with previous studies. While previous studies used few questions, this study confirms that women are deontological in the MI through a structured and validated questionnaire. Typical personal and impersonal moral dilemmas differ along two dimensions. Personal dilemmas are more emotionally salient and violate Kant's practical imperative that humans must never be used as mere means but only as ends. Capraro and Sippel [33] suggest that gender differences in these types of dilemmas are driven by emotional salience. If so, women may make more utilitarian judgments because the MI is a moral dilemma that rarely entails emotion. This may indicate that women are inherently more likely to make utilitarian judgments if the situation does not involve emotion.

In this study, we observed a significant trend toward not making utilitarian choices among participants in their 50s compared with those in their 20s in the personal moral dilemma task. Previous studies show that older adults tend to make more deontological moral judgments than younger adults [34]. Research suggests that older adults prioritize goals of maintaining positive experiences and distancing from negative experiences [35]. Therefore, older adults may be making deontological choices because making utilitarian choices in the personal moral dilemma task is a negative experience. The dual process model is apparently affected by emotion and causal reasoning [36]. Although previous studies have not confirmed that emotions change in adulthood with aging, it has been noted that reasoning declines linearly with age [37]. Therefore, the dual-process model may be affected by aging. Consequently, older adults may not have made utilitarian choices in the MP, where emotions have a greater effect, because the effects of emotions increase with age.

This study has certain limitations. First, in this study, a Japanese version was developed using Greene et al.'s [7] original version—which is the most commonly used moral dilemma task in research—and its reliability and validity were verified. Recently, a simplified version of the original version was developed [38], and it may be worthwhile to develop a Japanese version of the simplified version in the future. Second, the criterion validity testing in the MI revealed a weak positive correlation. However, the *p*-value was very small, indicating that a certain degree of the criterion validity in the MI was confirmed. Therefore, in the future, it may be beneficial to include MI items in the moral dilemma task questionnaire. Third, regarding regional differences, the sample size was small. Thus, future research with a larger sample and diverse types of moral judgments and behavioral intentions should be conducted to verify regional differences.

## 5. Conclusions

The Japanese version of Greene's moral dilemma task was reliable and valid. Furthermore, no regional differences were found within Japan, suggesting the possibility of obtaining common results across diverse cultures. However, gender differences were noted in moral judgments, with women and men tending to be more utilitarian in situations where emotions were less and more likely to be involved, respectively. The association between aging and deontological moral judgments seemed to support the theory of moral development.

**Supplementary Materials:** The following supporting information can be downloaded at: https://www.mdpi.com/article/10.3390/psych5010017/s1, Greene's Moral Dilemma Questionnaire: Japanese version.

**Author Contributions:** Conceptualization, Y.T. and A.Y.; methodology, Y.T. and A.Y.; formal analysis, A.Y.; writing—original draft preparation, A.Y.; writing—review and editing, Y.T.; funding acquisition, Y.T. All authors have read and agreed to the published version of the manuscript.

**Funding:** This research was supported by AMED, grant number SRPBS 16769942.

**Institutional Review Board Statement:** All procedures followed were in accordance with the ethical standards of the responsible committee on human experimentation (institutional and national). This study was conducted with the approval of the Research Ethics Review Committee of Graduate School of Medicine (review number: 11486-(1), 14 April 2017).

**Informed Consent Statement:** Informed consent was obtained from all patients included in the study.

**Data Availability Statement:** The data that support the findings of this study are available from the corresponding author upon reasonable request.

**Conflicts of Interest:** The authors declare no conflict of interest.

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
