# Peer review of "Verification of the Japanese Version of Greene’s Moral Dilemma Task’s Validity and Reliability"

_psych, doi:10.3390/psych5010017_

Round 1

Reviewer 1 Report

The article is interesting, and as proposals for improvement, they would be:

In the introduction there is a lot of physiological theory, which is then not used in the rest of the article. clarify.

Clarify the differences between point 2, 3, :"The response format was a four-point 104 scale: 1 = not appropriate, 2 = somewhat appropriate, 3 = somewhat appropriate, and 4 = appropriate".

Add an annex with the questionnaire.

In the analyzes, indicate the n of each sample, so that the comparisons are clearer.

Thank you.

Author Response

Reply to Reviewer 1

Thank you for your valuable suggestions on my manuscript.

#1 In the introduction there is a lot of physiological theory, which is then not used in the rest of the article. clarify.

 Thank you for your comment. We have added a discussion of the results from the dual-processing theory perspective for System 1 and System 2, which is the primary reason the physiological theory is discussed in the Introduction section. (Lines 255–257)

#2 Clarify the differences between point 2, 3, :"The response format was a four-point 104 scale: 1 = not appropriate, 2 = somewhat appropriate, 3 = somewhat appropriate, and 4 = appropriate".

Thank you for your comment. Accordingly, we have clarified the difference between points 2 and 3. (Lines 104–106)

#3 Add an annex with the questionnaire.

Thank you for your suggestion. Per your suggestion, we have attached, as an appendix, the questionnaire that we developed.

#4 In the analyzes, indicate the n of each sample, so that the comparisons are clearer.

Per your point, we have added the n of each sample in Tables 2 and 3.

Reviewer 2 Report

Thank you for allowing me to review your paper. I found the introduction, method and the results section very clear! However I have couple of suggestions for the discussion. First, the first sentence should be changed as a description of a) which were your questions/research rationale, b) what you have done to address the issue, and c) which is the main result of your study. Second, I don't see the point of sentence in the line 236. I think you should delete "The authors held a discussion regarding content validity" .Third, the limitations appear very destroying for your work. I suggest you rephrase them in a way that the limitations do not destroy your work but rather as something that can be done in the near future.

Author Response

Reply to Reviewer 2

Thank you for your valuable suggestions on my manuscript.

#1 First, the first sentence should be changed as a description of a) which were your questions/research rationale, b) what you have done to address the issue, and c) which is the main result of your study.

Thank you for your helpful comment. Per your suggestion, we have revised the Discussion section such that the first paragraph now discusses the aims of the study, what was done to address them, and the main results of the study. (Lines 225–236)

#2 Second, I don't see the point of sentence in the line 236. I think you should delete "The authors held a discussion regarding content validity" .

Thank you for pointing this out. We deleted the sentence “The authors held a discussion regarding content validity.”

#3 Third, the limitations appear very destroying for your work. I suggest you rephrase them in a way that the limitations do not destroy your work but rather as something that can be done in the near future.

Thank you for your suggestion. We agree with your opinion. We have revised the limitations to include suggestions for future studies so that it is positive and does not detract from the significance of this study’s findings. (Lines 298–308)